# STEERING MERGED LLMS FOR MULTILINGUAL REASONING WITH COEFFICIENT OPTIMIZATION

## ABSTRACT

Model merging has recently proven effective in enhancing the cross-lingual capabilities of reasoning models by integrating them with multilingual language models. However, existing methods typically apply a uniform merging strategy across languages, leading to a trade-off: while low-resource languages may benefit, high-resource languages such as English often suffer performance degradation. We attribute this limitation to insufficient coordination between the multilingual and reasoning models, where suboptimal representation merging impairs generalization. To mitigate this, we introduce **SteMerger**, a preference-driven framework that dynamically **steers** the model **merger** by optimizing the merging coefficients. Experiments on multilingual reasoning benchmarks show that SteMerger consistently improves performance across a wide range of languages, outperforming several strong baselines.

## 1 INTRODUCTION

Large language models (LLMs) have been shown to possess strong foundational reasoning abilities and have been successfully applied to complex tasks such as mathematical reasoning (Cobbe et al., 2021), commonsense reasoning (Patel et al., 2021), and natural language inference (Conneau et al., 2018). However, existing open-source reasoning LLMs, such as MetaMath (Yu et al., 2024) and MathOctopus (Chen et al., 2024b), still face significant challenges in non-English reasoning due to the imbalanced nature of multilingual pretraining data. This motivates research into **multilingual reasoning** (Shi et al., 2023; Chen et al., 2024b; She et al., 2024; Huang et al., 2024; Yoon et al., 2024; Bandarkar et al., 2025), which aims to extend the reasoning capabilities of LLMs to low-resource languages with limited supervision. While retraining-based methods may seem like straightforward solutions, such as translating query-response pairs and fine-tuning LLMs on them (Shi et al., 2023; Chen et al., 2024b), they incur high translation costs and often struggle due to insufficient translated data.

Alternatively, a series of model merging methods aim to compose the language and mathematical capabilities of multiple LLMs by combining an external multilingual language model that produces language-agnostic intermediate hidden states, which are then used to strengthen reasoning capabilities in low-resource languages (Yoon et al., 2024; Huang et al., 2024; Bandarkar et al., 2025). In particular, recent efforts have demonstrated that directly replacing specific network layers of the reasoning LLM with those from the multilingual LLM can enhance mathematical performance in the target language (Yoon et al., 2024; Bandarkar et al., 2025). During training, both the original multilingual model and the reasoning LLM are kept frozen, while a trainable mapping layer is introduced to align their representations due to inconsistency between the two latent spaces. Furthermore, Huang et al. (2024)

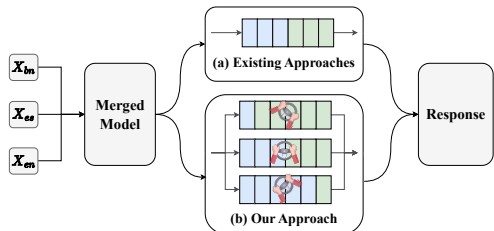

**Figure 1.** Conceptual illustration of our approach. Unlike existing approaches that adopt uniform representation merging across different instructions (top), **SteMerger** (bottom) learns to steer model behaviors by dynamically adjusting the merging coefficients.

resentations due to inconsistency between the two latent spaces. Furthermore, Huang et al. (2024)

introduces a two-stage mapping-augmentation scheme that collaboratively leverages both internal and external capabilities of LLMs, thereby preserving their core reasoning abilities.

However, existing merged models typically concatenate the representations from the multilingual model and the reasoning LLM, without considering the actual generation behavior. We observe that the representations extracted from the source models contribute differently to multilingual reasoning: the multilingual model offers strong text understanding for low-resource non-English languages, whereas the reasoning LLM provides robust mathematical reasoning for high-resource languages like English, which dominate pre-training data. This discrepancy suggests that relying solely on a uniform merged representation may fail to accurately capture the intended generation behavior. As a result, it often leads to suboptimal performance and occasional failures, particularly when the alignment between the multilingual and reasoning representations is inadequate.

To fill this gap, inspired by recent preference optimization techniques such as Direct Preference Optimization (DPO) (Rafailov et al., 2023), we propose an approach to steer the merged LLMs (**SteMerger**) via coefficient optimization. Instead of relying on a fixed concatenation of representations, our method enables the model to dynamically modulate the contribution of each source model (i.e., the multilingual language model and the reasoning LLM), allowing for more flexible and adaptive coordination between them. This design facilitates input-aware preference modeling, enabling the merged model to shift its inductive bias toward the source most aligned with the current input. As a result, it yields more accurate and targeted reasoning across diverse linguistic contexts. The optimized source models are not only individually improved but also serve as better generators of preference pairs, enabling the collection of higher-quality training data. This facilitates a new round of collaborative training, in which the merged model is guided by increasingly reliable supervision toward better coordination.

The main contributions of this paper are as follows:

- We introduce a model merging framework, **SteMerger**, designed to address the challenges of multilingual reasoning. By optimizing two adapters in different models, SteMerger dynamically coordinates the merging process based on the input query.
- We propose a coefficient optimization method, motivated by the observation that the performance of the merged model is sensitive to the variation in merging weights.
- Extensive experiments on multiple multilingual reasoning tasks demonstrate that SteMerger effectively enhances the coordination between the multilingual model and the reasoning LLM. It consistently outperforms baselines, achieving superior performance across diverse tasks.

## 2 RELATED WORK

### 2.1 MODEL MERGING

Model merging aims to combine the strengths of multiple models into a unified architecture and has been widely used to enhance capabilities such as modality integration (Sung et al., 2023; Chen et al., 2024a) and task generalization (Bandarkar et al., 2025; Du et al., 2025). Existing works can be broadly categorized into two types: homogeneous merging, which combines models with the same architecture, and heterogeneous merging, which merges models across architectural or modality boundaries. Recent studies have explored model merging for cross-lingual transfer learning (Yoon et al., 2024; Huang et al., 2024), but often suffer from limited controllability and alignment issues in multilingual reasoning settings. In contrast, our work introduces a preference-driven merging approach that dynamically steers the composition of multilingual representations, enabling better coordination between source models and improving generalization across both low-resource and high-resource languages.

### 2.2 PREFERENCE OPTIMIZATION

Preference optimization aims to adjust model behavior based on comparisons between generated outputs. A widely adopted paradigm is Reinforcement Learning from Human Feedback (RLHF), which learns a reward model from pairwise preferences (e.g., using the Bradley-Terry

model (Bradley & Terry, 1952)) and fine-tunes the base model to maximize expected rewards using reinforcement learning algorithms such as Proximal Policy Optimization (PPO) (Schulman et al., 2017). Despite its effectiveness, RLHF introduces additional complexity due to the need for explicit reward modeling. Recent advances like Direct Preference Optimization (DPO) (Rafailov et al., 2023) bypass the reward modeling stage by directly fitting the model to preference data. This not only simplifies the pipeline but also aligns more closely with the true objective of preference learning. Inspired by this, we adopt a preference-driven optimization strategy to steer the reasoning behavior of merged LLMs. Instead of updating the full model, we introduce lightweight residual adapters trained via DPO, enabling fine-grained control over representation merging.

## 3 Analysing Current Merged Models for Multilingual Reasoning

In this section, we investigate how multilingual reasoning performance is influenced by the coordination of representations in the merged model. Specifically, we focus on the impact of merging two distinct source models, e.g. one trained for multilingual understanding and the other for mathematical reasoning, on the performance across low-resource and high-resource languages.

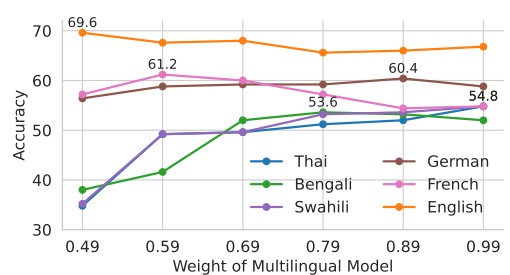

**Figure 2.** Multilingual reasoning results with different multilingual model weights.

Although prior work (Yoon et al., 2024; Huang et al., 2024) has demonstrated substantial gains in low-resource language reasoning via model merging, a notable performance gap remains. In particular, merged models may underperform compared to the original reasoning model on high-resource languages. For instance, LangBridge reports a drop of over 2 accuracy points on English (Yoon et al., 2024).

This raises an important question: *What factors govern the trade-off between gains in low-resource languages and degradation in high-resource ones during merging?* We hypothesize that the key lies in the coordination of the merged representations, which we refer to as the **representation merging coefficients**. Specifically, these coefficients control the relative contributions of the multilingual model (e.g., mT5) and the reasoning model (e.g., MetaMath) during merging.

To verify this hypothesis, we conduct a quantitative analysis using a static gating mechanism (Sung et al., 2023), which assigns a fixed weight to the multilingual representation during inference. A higher weight implies greater reliance on the multilingual features within the merged representation.

As shown in Figure 2, we observe that increasing the multilingual weight improves performance on low-resource languages, while performance on high-resource languages often declines. This indicates a trade-off driven by how the merged model balances cross-lingual generalization and reasoning ability. Moreover, we compute the Pearson correlation between the multilingual model weight and accuracy across languages. The strong correlation (Table 1) further confirms that multilingual reasoning performance is tightly linked to the coordination between the two source models.

**Table 1.** Pearson correlation coefficient $r$ between multilingual model weight and MGSM accuracy.

| Lang. | Th | Bn | Sw | De | Fr | En |
|---|---|---|---|---|---|---|
| $r$ | 0.84 | 0.83 | 0.85 | 0.68 | -0.69 | -0.77 |

## 4 SteMerger: Steering Merged Models via Coefficient Optimization

Based on the aforementioned analysis, we believe that **optimizing** the representation merging coefficients is key to improving multilingual reasoning. In this section, we propose **SteMerger**, a preference-driven framework for multilingual reasoning that dynamically coordinates the integra-

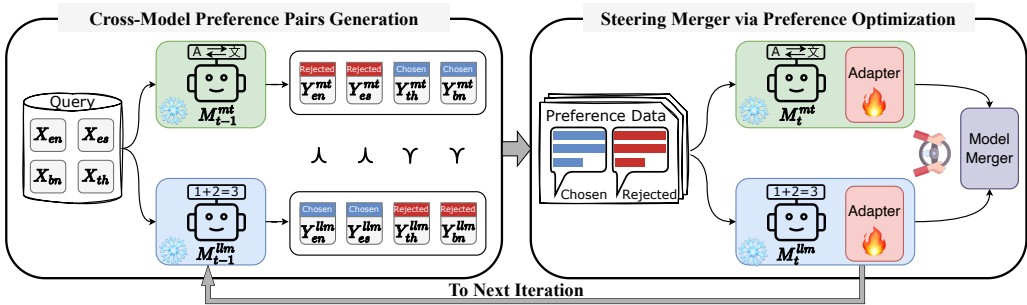

**Figure 3.** Overview of **SteMerger**. Our method iteratively merges the capabilities of a multilingual model and a reasoning model through preference-driven training. Each iteration consists of two steps: (i) *Preference pair generation*: For each multilingual query, responses are generated independently using both source models. Their outputs are compared to derive preference pairs, reflecting which model performs better for each query. (ii) *Steering models via preference optimization*: The constructed preference pairs are used to steer the merged model via a combined DPO and NLL objective. The resulting merged model $\mathcal{M}_t$ (consists of $M_t^{mt}$ and $M_t^{llm}$) inherits the strengths of both sources and is used in the next iteration.

tion of a multilingual model and a reasoning LLM. Our method builds upon prior model merging frameworks (Yoon et al., 2024; Huang et al., 2024), which integrate a multilingual model $M^{mt}$ and a reasoning LLM $M^{llm}$ through a trainable mapping layer. For clarity, we denote the merged model as $\mathcal{M}$ to distinguish it from the two individual source models $M^{mt}$ and $M^{llm}$.

Unlike prior approaches that apply uniform merging strategies, **SteMerger** aims to adaptively adjust the merging behavior based on specific instructions, enabling better coordination between multilingual understanding and reasoning precision. To this end, we adopt an iterative training framework, where each iteration consists of two stages: (i) multilingual preference pair generation and (ii) steering the merger via preference optimization, as illustrated in Figure 3. At iteration $t$, we use the current models $M_{t-1}^{mt}$ and $M_{t-1}^{llm}$ to generate multilingual preference data, which is then used to optimize a lightweight steering module via Direct Preference Optimization (DPO). The resulting merged model $\mathcal{M}_t$, consisting of $M_t^{mt}$ and $M_t^{llm}$, is then used to initialize the next iteration. This loop enables the model to continually refine its multilingual alignment through preference-driven learning.

**Initialization of Model Merger** Given a query $q$ consisting of $l$ tokens, a pre-trained multilingual model first generates a semantic representation $X \in \mathbb{R}^{l \times d_1}$, which captures cross-lingual understanding. Since the multilingual model and the reasoning LLM reside in different representation spaces, a mapping layer (e.g., a two-layer MLP) is used to project $X$ into the LLM space, yielding $X_f \in \mathbb{R}^{l \times d_2}$.

To leverage both external and internal knowledge, the merged model also computes the native LLM embedding $T \in \mathbb{R}^{l \times d_2}$ of the same input (Huang et al., 2024). These two representations are then concatenated with special tokens to form a unified input: $[\texttt{<bos>}; X_f; \texttt{<sep>}; T]$, which is passed to the LLM for response generation. During training, the encoder and LLM remain frozen, and only the mapping components and boundary tokens are updated.

### 4.1 Cross-Model Preference Pairs Generation

To construct reliable preference data that reflects the comparative strengths of multiple source models, we adopt a response-level comparison strategy. Given an instruction $x_i$, we first obtain responses independently from the multilingual language models $M^{mt}$ and reasoning LLM $M^{llm}$, resulting in $y_i^{mt} \sim M^{mt}(x_i)$ and $y_i^{llm} \sim M^{llm}(x_i)$. During this process, each model uses its own representation space, allowing it to fully leverage its internal capabilities without being constrained by a unified embedding space.

The two responses are then compared based on task label correctness to determine the preference: the better response is labeled as the *winner* and the other as the *loser*. This yields a pairwise preference triple $(x_i, y_i^{\text{win}}, y_i^{\text{lose}})$, which is used to supervise downstream coefficient optimization.

However, in practice, we observe that many instruction fall into one of two ambiguous categories: (i) both models provide equally correct responses, or (ii) both fail to generate valid outputs. Including such instances in preference training can introduce noise or misaligned preference signals. Therefore, we explicitly discard samples where both responses are of similar quality, as these do not offer clear preference supervision.

To avoid wasting these filtered examples, we instead utilize them under standard supervised learning: if both responses are acceptable, we randomly select one and apply a negative log-likelihood (NLL) loss against the reference answer. This allows the model to still benefit from correct supervision even in the absence of a preference signal. The final objective for these samples is:

$$\mathcal{L}_{\text{NLL}} = -\mathbb{E}_{(x,y^*) \sim \mathcal{D}_{\text{neutral}}} \frac{1}{|y^*|} \log \mathcal{M}_\theta(y^* \mid x) \tag{1}$$

where $\mathcal{D}_{\text{neutral}}$ denotes the filtered examples with no clear win/lose label, and $y^*$ is the selected valid response (or reference label, if available). This hybrid approach ensures full utilization of training data while maintaining the integrity of preference optimization.

## 4.2 STEERING MERGER VIA COEFFICIENT OPTIMIZATION

To enable fine-grained control over the behavior of model merger, we introduce a steering merger process consisting of lightweight, trainable adapters using residual connection. Rather than retraining the entire model architecture, these adapters are optimized to reflect the preferences embedded in the source models, offering a flexible and efficient way to align the model outputs with desirable traits from each constituent source.

We cast merger steering as a preference optimization problem, where pairwise preference data extracted from the outputs of two source models guides the coefficients optimization process. To this end, we apply contrastive-based loss functions to iteratively adjust the adapters. This optimization reinforces behaviors that are favored by the source models while suppressing less desirable ones, ultimately improving the multilingual generalization of the merged model with various reasoning tasks.

More concretely, given a multilingual preference triplet $\{x, y^{\text{win}}, y^{\text{lose}}\} \in \mathcal{D}$, we adopt a variant of direct preference optimization (DPO) to encourage the generation of preferred responses. Our formulation enhances the standard DPO objective (Rafailov et al., 2023) by incorporating the NLL loss Equation (1), particularly in multilingual settings. The full loss is defined as:

$$\mathcal{L}_{\text{DPO}} = -\mathbb{E}_{(x,y^{\text{win}},y^{\text{lose}}) \sim \mathcal{D}} \log \sigma \left( \beta \left[ \log \frac{\mathcal{M}_\theta(y^{\text{win}} \mid x)}{\mathcal{M}_{\text{ref}}(y^{\text{win}} \mid x)} - \log \frac{\mathcal{M}_\theta(y^{\text{lose}} \mid x)}{\mathcal{M}_{\text{ref}}(y^{\text{lose}} \mid x)} \right] \right) \tag{2}$$

$$\mathcal{L} = \mathcal{L}_{\text{DPO}} + \alpha \cdot \mathcal{L}_{\text{NLL}} \tag{3}$$

Here, $\mathcal{M}_\theta(\cdot \mid x)$ denotes the trainable policy model, and $\mathcal{M}_{\text{ref}}(\cdot \mid x)$ is the frozen reference model. The steering parameters $\theta$ are initialized from the reference and iteratively updated. The sigmoid function $\sigma$ governs the preference margin, and $\alpha$ balances the DPO loss and NLL loss. During training, only the adapter parameters are updated, while all parameters of the pretrained backbone models remain frozen.

**Iterative Optimization** We propose an iterative training framework that progressively refines a base instruction-following model through multilingual preference optimization. Starting from an initial merged model $\mathcal{M}_0$, we generate a sequence of models $\mathcal{M}_1, \mathcal{M}_2, \ldots, \mathcal{M}_T$, where each iteration incorporates newly generated preference data to enhance performance.

**Table 2.** Accuracy (%) results on MGSM. Lrl., Hrl., and Avg. represent the average accuracy across low-resource languages, high-resource languages, and all languages, respectively. We regard Bn, Th, and Sw as low-resourse languages, and regard the remaining languages as high-resource languages. The best performance is in bold (same for Table 3 and Table 4).

| MGSM | Bn | Th | Sw | Ja | Zh | De | Fr | Ru | Es | En | Lrl. | Hrl. | Avg. |
|---|---|---|---|---|---|---|---|---|---|---|---|---|---|
| | | | | | Original PLMs | | | | | | | | |
| MetaMath [2024] | 6.8 | 7.2 | 6.8 | 36.4 | 38.4 | 55.2 | 54.4 | 52.0 | 57.2 | 68.8 | 6.9 | 51.8 | 38.3 |
| | | | | | Baselines | | | | | | | | |
| MultiReason [2024] | 33.2 | 40.0 | 42.0 | 42.0 | 42.0 | 45.2 | 44.8 | 45.2 | 48.0 | 52.0 | 38.4 | 45.6 | 43.4 |
| QAlign [2024] | 32.4 | 39.6 | 40.4 | 44.0 | 48.4 | 54.8 | 56.8 | 52.4 | 59.6 | 68.0 | 37.5 | 54.9 | 49.6 |
| LangBridge [2024] | 42.8 | 50.4 | 43.2 | 40.0 | 45.2 | 56.4 | 50.8 | 52.4 | 58.0 | 63.2 | 45.5 | 52.3 | 50.2 |
| Translate-En [2023] | 48.4 | 37.6 | 37.6 | 49.2 | 46.8 | 60.4 | 56.4 | 47.6 | 59.6 | 65.5 | 41.2 | 55.1 | 50.6 |
| MindMerger [2024] | 50.4 | 52.8 | 57.2 | 54.4 | 53.6 | 61.2 | 57.6 | 60.8 | 58.4 | 66.8 | 53.5 | 59.0 | 57.3 |
| | | | | | Our Method | | | | | | | | |
| Merger* | 50.5 | 52.9 | **57.6** | 52.8 | 54.8 | 59.1 | 56.8 | 60.7 | 61.7 | 66.7 | 53.7 | 58.9 | 57.4 |
| **SteMerger** | **52.0** | **54.5** | 56.3 | **53.5** | **55.9** | **60.0** | **60.5** | **61.3** | **62.8** | **68.0** | **54.3** | **60.3** | **58.5** |

**Table 3.** Accuracy (%) on X-CSQA. Avg. represents the average accuracy across all languages.

| X-CSQA | Sw | Ur | Hi | Ar | Vi | Ja | Pl | Zh | Nl | Ru | It | De | Pt | Fr | Es | En | Avg. |
|---|---|---|---|---|---|---|---|---|---|---|---|---|---|---|---|---|---|
| | | | | | | Original PLMs | | | | | | | | | | | |
| MetaMath [2024] | 24.2 | 25.1 | 32.9 | 32.3 | 50.9 | 49.1 | 50.6 | 56.5 | 57.5 | 56.0 | 56.0 | 61.2 | 61.7 | 63.5 | 64.0 | 76.3 | 51.3 |
| | | | | | | Baselines | | | | | | | | | | | |
| MultiReason [2024] | 27.6 | 29.2 | 32.0 | 28.7 | 38.8 | 38.7 | 45.5 | 43.8 | 45.9 | 46.5 | 50.2 | 49.1 | 51.2 | 52.1 | 54.3 | 67.2 | 43.8 |
| QAlign [2024] | 35.1 | 32.6 | 37.8 | 36.3 | 50.5 | 49.2 | 51.3 | 54.8 | 56.3 | 56.3 | 58.3 | 58.8 | 59.8 | 60.3 | 63.1 | 75.7 | 52.3 |
| LangBridge [2024] | 31.8 | 30.5 | 30.6 | 30.6 | 33.3 | 33.9 | 39.8 | 39.8 | 38.4 | 35.1 | 39.1 | 37.4 | 36.3 | 38.2 | 38.4 | 44.4 | 36.1 |
| Translate-En [2023] | 36.5 | 41.3 | 48.4 | 44.6 | 51.8 | 47.1 | 53.3 | 51.5 | 55.0 | 54.4 | 56.3 | 57.3 | 54.7 | 57.2 | 55.5 | 71.3 | 52.3 |
| MindMerger [2024] | 45.5 | 46.2 | 48.4 | 51.4 | 60.6 | 53.9 | 63.3 | 62.9 | 63.8 | **63.7** | **66.8** | **67.0** | **67.1** | **68.1** | **69.1** | **78.1** | 61.0 |
| | | | | | | Our Method | | | | | | | | | | | |
| Merger* | 46.9 | 52.0 | 50.4 | **55.7** | 59.5 | **56.5** | 64.2 | 62.8 | 63.5 | 62.4 | 65.1 | 64.4 | 65.9 | 65.2 | 66.6 | 76.3 | 61.1 |
| **SteMerger** | **48.1** | **52.9** | **50.6** | 55.4 | **60.9** | 56.4 | **64.8** | **63.2** | **63.9** | 62.1 | 65.3 | 64.2 | 66.0 | 64.9 | 67.0 | 75.3 | **61.3** |

# 5 EXPERIMENTS

This section first introduces the multilingual reasoning benchmark, followed by a brief description of the experimental configurations and baseline models. Finally, we present the main results of both the baselines and SteMerger across three datasets.

## 5.1 EVALUATION DATASETS

We categorize our experiments into the following three task types: **(1)** Mathematical reasoning. We evaluate on two multilingual math word problem datasets in this category. **MGSM** (Shi et al., 2023) consists of grade-school level math questions translated by humans into 11 typologically diverse languages. The original English prompts are sampled from GSM8K (Cobbe et al., 2021), a benchmark designed to test step-by-step arithmetic reasoning. **(2)** Commonsense reasoning. We use **X-CSQA** (Lin et al., 2021), a multilingual extension of the CommonsenseQA dataset. The original CSQA task is a multiple-choice question-answering benchmark targeting general commonsense knowledge, but it is available only in English. X-CSQA provides translated versions of CSQA across multiple languages, along with a new data split to support cross-lingual evaluation. The dataset includes 8,888 English training examples, 1,000 development examples per language, and 1,074 test examples per language. **(3)** Natural language inference. We evaluate natural language inference using **XNLI** (Conneau et al., 2018), a widely used multilingual benchmark spanning 15 languages. The task involves determining whether a given *hypothesis* logically follows from a *premise*, categorized as entailment, contradiction, or neutral. The dataset covers languages both typologically close to English (e.g., French, German, Spanish) and more distant (e.g., Arabic, Thai, Swahili), making it well-suited for evaluating cross-lingual generalization.

**Table 4.** Accuracy (%) on XNLI. Avg. represents the average accuracy across all languages.

| XNLI | Sw | Ur | Hi | Th | Ar | Tr | El | Vi | Zh | Ru | Bg | De | Fr | Es | En | Avg. |
|---|---|---|---|---|---|---|---|---|---|---|---|---|---|---|---|---|
| Original PLMs | | | | | | | | | | | | | | | | |
| MetaMath [2024] | 45.9 | 49.2 | 55.7 | 55.4 | 60.9 | 61.9 | 63.7 | 73.7 | 74.7 | 77.6 | 76.7 | 80.6 | 82.2 | 82.8 | 90.0 | 68.7 |
| Baselines | | | | | | | | | | | | | | | | |
| MultiReason [2024] | 56.3 | 57.5 | 61.7 | 60.1 | 61.7 | 65.6 | 67.0 | 73.7 | 79.1 | 79.7 | 78.7 | 82.3 | 82.9 | 83.9 | 88.8 | 71.9 |
| QAlign [2024] | 65.2 | 62.2 | 63.3 | 65.2 | 67.0 | 67.9 | 66.5 | 73.7 | 76.6 | 79.2 | 79.4 | 80.9 | 83.1 | 83.8 | 89.1 | 73.5 |
| LangBridge [2024] | 71.7 | 66.9 | 71.1 | 72.4 | 75.2 | 74.8 | 79.1 | 78.5 | 77.4 | 77.4 | 79.6 | 78.8 | 79.9 | 80.5 | 83.4 | 76.5 |
| Translate-En [2023] | 65.3 | 61.6 | 68.7 | 69.5 | 68.9 | 74.5 | 79.3 | 76.7 | 74.8 | 76.0 | 80.8 | 80.6 | 80.4 | 81.4 | 87.4 | 75.1 |
| MindMerger [2024] | 66.6 | 69.4 | 74.7 | 71.8 | 76.2 | 75.7 | 78.5 | **80.3** | 80.0 | 80.7 | 82.4 | **83.5** | 83.9 | 84.4 | 88.7 | 78.4 |
| Our Method | | | | | | | | | | | | | | | | |
| Merger* | 72.2 | 68.8 | 72.7 | 72.8 | 75.8 | 76.6 | 77.7 | 79.2 | 80.4 | 80.8 | 82.4 | 82.6 | 84.0 | 83.9 | 88.6 | 78.6 |
| **SteMerger** | **72.7** | **71.8** | **75.1** | **74.2** | **77.6** | **77.2** | **80.0** | 80.1 | **81.0** | **82.2** | **83.3** | 83.5 | **84.7** | **84.7** | **88.9** | **79.8** |

## 5.2 BASELINES

To evaluate the generality and effectiveness of our approach, we conduct experiments using the strong reasoning LLM **MetaMath** (Yu et al., 2024), which serves as the backbone architectures for all baseline methods except QAlign. We compare our method against several state-of-the-art baselines for multilingual reasoning: **(1) MultiReason** (Zhu et al., 2024): A method that enhances reasoning consistency across languages via question alignment and rationale generation. **(2) QAlign** (Zhu et al., 2024): A framework that aligns questions across languages through fine-tuned translation-based contrastive learning. **(3) LangBridge** (Yoon et al., 2024): A cross-lingual transfer technique that bridges non-English inputs to an English-centric reasoning space. **(4) Translate-En** (Shi et al., 2023): A simple but effective approach that translates non-English inputs to English and uses an English reasoning model. **(5) MindMerger** (Huang et al., 2024): A multilingual method that merges task representations across languages to promote cross-lingual reasoning alignment.

## 5.3 IMPLEMENTATION DETAILS

During the initialization stage of model merging, we follow the MindMerger approach and make use of the translation training data released by the authors. In the preference optimization stage, no additional data are introduced; instead, the preference training samples are constructed from the available training data as described in the methodology section. For a fair comparison with prior work (Yoon et al., 2024; Huang et al., 2024), we adopt the encoder of mT5-xl (Xue et al., 2021) as the multilingual backbone in our methods, and employ LLaMA 2-7B (Touvron et al., 2023) as the large language model across all experiments. The final model is selected based on the averaged performance of all languages on the dev set. We perform grid search over the balance training parameter $\alpha$ and learning rate from [0.1, 0.2] and [1e-3, 1e-4, 2e-5]. We conduct each experiment 3 times with different random seeds and report the average results of 3 run experiments [1]. Given the availability of translated training data, we choose a mid-resource source language (e.g., Chinese) for coefficient optimization to balance generalization.

## 5.4 MAIN RESULTS

**(1) SteMerger consistently outperforms all baselines across diverse reasoning tasks.** We evaluate our proposed **SteMerger** model on three reasoning benchmarks spanning arithmetic (MGSM), commonsense (X-CSQA), and natural language inference (XNLI). As shown in Table 2, Table 3, and Table 4, **SteMerger** achieves consistent performance gains across all tasks, demonstrating strong generalization. The performance gains on the X-CSQA are relatively limited. We guess that the limited gains stem from their discrete choice-format outputs, which provide weak preference signals and hinder effective preference optimization.

**(2) SteMerger surpasses strong baselines under identical training and decoding conditions.** Compared to existing strong multilingual reasoning baselines such as **MultiReason** and **Mind-**

---

[1]We used the checkpoints given by MindMerger (Huang et al., 2024) as the initial Model Merger.

**Merger**, our method achieves superior performance under identical conditions (same prompts and training data). Our approach can be used as a plug-in on top of standard decoding strategies, yielding average gains of +1.3%, +0.2%, and +1.5% over the best baseline on the three tasks, respectively.

**(3) SteMerger generalizes effectively to both low-resource and high-resource languages.** Notably, our method consistently improves performance across all languages. As shown in Table 2, compared to the initial merger model, **SteMerger** achieves an average gain of +0.6% on low-resource languages and +1.4% on high-resource ones. In particular, we observe substantial improvements on Thai (+1.6%) and French (+1.7%), demonstrating the model to preserve and enhance task-specific reasoning across languages with varying resource levels.

## 6 ANALYSIS AND DISCUSSION

To better understand SteMerger and explore how SteMerger influences the multilingual reasoning, we conduct analyses on several questions. Results show SteMerger achieves performance improvements across different languages and improves the representation merging coefficients.

**(Q1) How SteMerger influences merging representation consistency?**

We compute the cosine similarity between the target language and English in the merged representation space. As shown in Figure 4, compared to the initial merger, SteMerger consistently improves cross-lingual alignment in both low-resource and high-resource languages. To further assess the impact of representation consistency on reasoning performance, we calculate the Pearson correlation between cosine similarity and task accuracy on MGSM. The correlation coefficient is $r = 0.8045$ with $p = 0.0050$, indicating a statistically significant positive relationship between representation consistency and reasoning accuracy. The results demonstrate the effectiveness of SteMerger for multilingual reasoning.

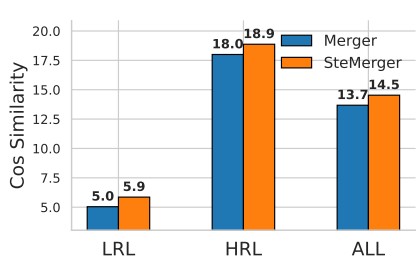

**Figure 4.** Comparing the representation similarity between English and target languages.

**(Q2) How SteMerger influences multilingual prediction answer consistency?**

Another advantage of SteMerger is the improvement it brings in the consistency of predicted answers across multilingual queries. This refers to a higher degree of agreement in responses to the same question posed in different languages. As shown in Figure 5, SteMerger outperforms the vanilla merger in terms of cross-lingual answer consistency, demonstrating greater stability across languages. This effect is particularly pronounced in typologically distant languages such as Bengali, Thai, and Swahili, where inconsistent reasoning is more common. These results highlight the effectiveness of SteMerger in transferring reasoning capabilities from English to other languages, and strengthen the model to handle multilingual queries in a consistent and reliable manner.

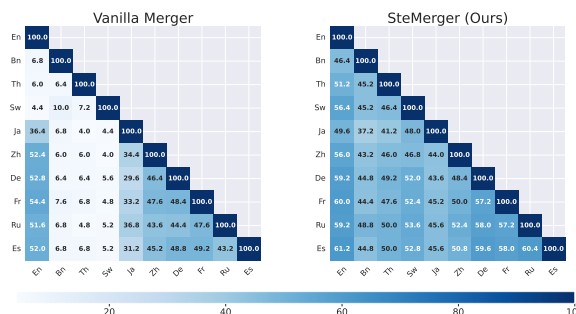

**Figure 5.** Comparing the prediction consistency of different models. Darker blue denotes higher level of prediction consistency.

**(Q3) What is the essential component of SteMerger?**

We conduct an ablation study, as shown in Table 5. First, we assess the impact of iterative preference optimization. The variant *w/o Iteration* applies preference learning only once to the initial Merger. The performance gap between SteMerger and *w/o Iteration* highlights the benefits of incorporating additional preference signals across multiple rounds, confirming the advantage of iterative refine-

**Table 5.** Ablation results on MGSM, where *w/o iteration* denotes remove the multiple rounds and inference and *w/ MultiSource* denotes using multiple source training data.

| MGSM | Bn | Th | Sw | Ja | Zh | De | Fr | Ru | Es | En | Lrl. | Hrl. | Avg. |
|---|---|---|---|---|---|---|---|---|---|---|---|---|---|
| **SteMerger** | 52.0 | **54.5** | 56.3 | **53.5** | **55.9** | 60.0 | 60.5 | 61.3 | 62.8 | **68.0** | 54.3 | **60.3** | 58.5 |
| *w/o Iteration* | 52.4 | 52.3 | 55.6 | 53.2 | 53.6 | **60.8** | 60.0 | 62.0 | **63.1** | 66.8 | 53.4 | 59.9 | 58.0 |
| *w/o NLL Loss* | 50.4 | 52.8 | 56.8 | 50.8 | 54.0 | 58.8 | 59.2 | **62.8** | 62.0 | 65.6 | 53.3 | 59.1 | 57.3 |
| *w/ MultiSource* | **54.4** | 53.2 | **59.2** | 52.0 | 53.6 | 59.6 | **61.6** | **62.8** | 62.8 | 66.8 | **55.6** | 59.9 | **58.6** |

ment. Second, *w/o NLL Loss* demonstrates greater effectiveness than relying solely on preference optimization loss, as it enables the model to learn from non-preferred examples through likelihood-based supervision. Finally, the *w/ MultiSource* variant incorporates preference data from multiple source languages (e.g., Chinese, German, Japanese) during training. Compared to SteMerger trained on a single source language (e.g., Chinese), this configuration improves performance on some target languages while degrading it on others. We hypothesize that although incorporating diverse sources enhances generalization, it may also introduce increased parameter interference, which can negatively affect model consistency and stability.

### (Q4) Which languages are most effective for SteMerger in multilingual reasoning?

We evaluate the performance of SteMerger using different source languages on MGSM, testing across ten languages that include both low-resource and high-resource cases. In the left and middle subplots of Figure 6, Bengali (bn) and English (en) achieve the highest average accuracy on low-resource and high-resource languages, respectively. This suggests that while SteMerger enhances the coordination between the multilingual model and the reasoning LLM, its effectiveness is still influenced by inherent biases present in the training data. Nonetheless, as shown in the right subplot of Figure 6, all variants outperform the initial merger baseline (57.4%) reported in Table 2, regardless of the training language. This demonstrates that the proposed coefficient optimization provides robust improvements, even when supervision is derived from a single language.

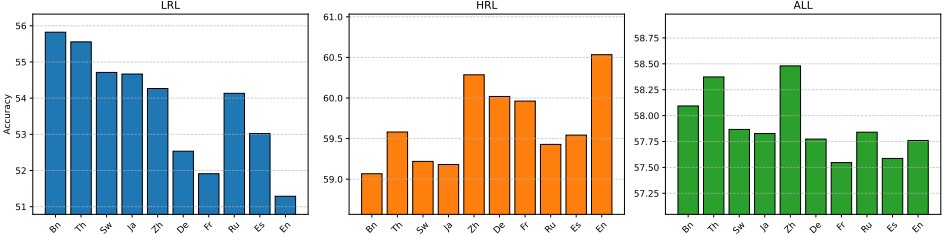

**Figure 6.** Results of SteMerger trained on data in different languages.

## 7 CONCLUSION

This work addresses a fundamental challenge in multilingual reasoning, how to effectively coordinate the strengths of multilingual and reasoning models through model merging. Our analysis reveals that static, uniform merging strategies can introduce a trade-off: while improving performance on low-resource languages, they often degrade reasoning in high-resource languages due to suboptimal representation coordination. To address this, we propose **SteMerger**, a lightweight, preference-driven framework that iteratively steers the merged model toward balanced multilingual reasoning via dynamic adjustment of representation merging coefficients. Experiments on three multilingual reasoning benchmarks demonstrate consistent gains across both high-resource and low-resource languages. Beyond performance, we further uncover a strong correlation between answer consistency and representation alignment, providing empirical insight into the mechanisms underlying multilingual generalization. Additionally, our findings suggest that dynamic input-aware merging strategies can serve as a promising direction for unifying multiple capabilities in large models.

ETHICS STATEMENT

This research focuses solely on general scientific tasks (i.e. multilingual reasoning) and involves no health, safety, privacy, or security risks. It does not involve human subjects, release new datasets, or introduce potentially harmful methods or applications. No issues related to legal compliance or research integrity are present. We affirm adherence to ethical standards throughout the study.

REPRODUCIBILITY STATEMENT

We provide a comprehensive description of the proposed SteMerger in section 4, with detailed implementation specifics provided in section 5. All datasets utilized in this research are publicly available. And the complete code will be made publicly available upon acceptance of the paper.

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

## A  APPENDIX

### THE USE OF LLMS

In this work, large language models (LLMs) were used solely during the final stages of the writing process, specifically for tasks such as proofreading and improving linguistic clarity. Their use was limited to enhancing the readability, fluency, and grammatical correctness of the manuscript, ensuring the effective communication of our ideas. Crucially, LLMs were not involved in any core aspects of the research itself, including the formulation of methodology, experimental design, or result interpretation. We take full responsibility for the content presented in this paper.

