# OpenReview forum: "Steering Merged LLMs for Multilingual Reasoning with Coefficient Optimization"
_ICLR.cc/2026/Conference — ICLR 2026 Conference Withdrawn Submission_

### Official Review · Reviewer_Q1YD · 2025-10-17

**Soundness:** 2
**Presentation:** 2
**Contribution:** 2
**Rating:** 4
**Confidence:** 4

**Summary:**

This paper argues that there is a limitation of insufficient coordination between the multilingual and reasoning models in model merging, leading to suboptimal representation merging impairs generalisation. To address this issue, the authors introduce a preference-driven framework that dynamically steers the model merger by optimising the merging coefficients.

**Strengths:**

- This paper introduces a model merging framework for multilingual reasoning, which dynamically coordinates the merging process through optimising two adapters in different models.
- The authors conduct extensive experiments on three different reasoning tasks, demonstrating that the proposed method consistently outperforms several baselines.
- The analysis and observation on the trade-off between the multilingual model and the reasoning model in Section 3 is interesting, which might provide important insight for future research.

**Weaknesses:**

- Since the preference data is generated by the multilingual model and reasoning model, it is interesting to see the win-rate result between the models for each language, which can serve as a plus for Section 3
- In lines 267-269, it misses some important details about iteration, including: (1) is it online or offline generation? (2) how many iterations do you have? (3) the preference data is generated by the multilingual model and reasoning model in the first iteration, what about the following iterations?
- This paper argues that translation incurs a high cost, while the proposed method requires additional cost to generate preference data
- The experimental setting is somewhat unfair because the backbone comes from a baseline MindMerger. It would be more fair to have the same start point for all experiments
- The performance improvement is limited: (1) compared to the backbone, the performance gain of the proposed method is limited (the percentage point improvement of the three tasks is 1.2, 0.3, and 1.4, respectively); (2) English still has performance degradation (Table 3), which is the main problem in the previous work proposed in this paper

Others:
- Line 362: no space between footnote number and text
- What is the difference between Merger* and SteMerger?

**Questions:**

See above Weaknesses.

---

### Official Review · Reviewer_6VRg · 2025-10-27

**Soundness:** 3
**Presentation:** 3
**Contribution:** 2
**Rating:** 4
**Confidence:** 4

**Summary:**

The paper addresses a limitation in model merging approaches that combine multilingual and reasoning LLMs. Existing methods merge model representations uniformly across languages, which helps low-resource languages but often degrades performance in high-resource ones like English. This imbalance stems from the lack of coordination between multilingual and reasoning representations.

To solve this, the authors propose SteMerger, a preference-driven, coefficient-optimization framework that dynamically adjusts how much each source model (multilingual and reasoning) contributes during inference. Instead of fixed merging weights, SteMerger learns optimal coefficients via Direct Preference Optimization (DPO), using pairwise preferences between model outputs to guide training. This allows the merged model to “steer” its behavior based on input language or task context.

**Strengths:**

- Novel model-merging framework with optimized coefficients instead of fixed blending.
- Lightweight implementation via adapters.
- Balanced performance across low- and high-resource languages.

**Weaknesses:**

- The architectural novelty is limited and mainly extends existing merging and preference optimization methods.
- The approach strongly depends on the quality of generated preference data.
- There is no human evaluation of reasoning quality or linguistic fluency.
- The paper does not analyze scaling behavior across larger models.

**Questions:**

- Are there existing coefficient optimization methods (e.g., AdamMS) that inspired this approach?
- Can SteMerger generalize beyond multilingual reasoning, for instance, to multimodal tasks?
- How are preference pairs labeled when outputs differ subtly (e.g., semantically correct but numerically inaccurate)?
- How are low-resource and high-resource languages defined quantitatively?

---

### Official Review · Reviewer_rWdp · 2025-10-27

**Soundness:** 2
**Presentation:** 2
**Contribution:** 2
**Rating:** 2
**Confidence:** 3

**Summary:**

This paper introduces SteMerger, a framework that iteratively generates cross-model preference pairs by creating pairs of answers from a multilingual model (in this case, mT5-xl) and a reasoning model (based on llama2-7B), henceforth LLM. Then, preferences are generated by checking the correctness of each answer.

Both models are optimized via adapters using DPO (Rafailov et al., 2023) and a weighted supervised fine-tuning loss. The model merging is done via concatenating (1) representations from the multilingual model, mapped into LLM representation space via a two-layer MLP (trained) and (2) representations from the 'native' LLM, and passed to the LLM for response generation, only the mapper and boundary tokens are updated. The method can be done in multiple iterations of obtaining data and training.

This method is applied and tested on three multilingual datasets: MGSM (Shi et al., 2023), a math dataset; X-CSQA (Lin et al., 2021), a multiple-choice QA dataset; and XNLI (Conneau et al., 2018), a natural language inference dataset. The paper compares against six baselines and shows that SteMerger has higher performance.

In the ablations, (1) the paper investigates how the merging affects the representations of the target language compared to English in MGSM. The paper shows that using SteMerger yields higher cosine similarity. (2) The paper investigates answer consistency in MGSM (?) where it shows that SteMerger yields more consistent answers. (3) The paper shows that using training data from multiple sources improves performance and that leaving out components such as the SFT loss and iterative training degrades performance. (4) In the last analysis, the paper shows that SteMerger works best for Bengali and English.

**Strengths:**

- The paper depicts a relatively simple method to improve model merging for multilingual reasoning.
- The method improves upon baselines in the framing of the paper.
- The method is applied to a diverse set of tasks (math, MCQA, NLI).

**Weaknesses:**

- The writing can be improved. In many cases, there is guesswork left for the reader.
    - For example, in analysis Q2, it is investigated whether the proposed method provides consistent answers. (1) It is implied that the analysis is done on MGSM, but not explicitly written. (2) Over how many predictions is this calculated?
    - It is mentioned that the method can be applied for multiple iterations. How many worked best?
    - As the experiments have already been conducted over three random seeds, why is the standard deviation not shown or a statistical significance test being conducted in Table 2, 3, and 4? The differences in performance seem marginal. As in analysis Q1 the paper applied statistical significance testing, it is missing from the rest of the experiments.
-  The core method is weighting the supervised fine-tuning loss of the correct label (L225). The proposed range is [0.1, 0.2] (L362), why was this deemed the best? This analysis/ablation seems critical but is missing from the paper.

**Questions:**

- Typological diversity in languages is a laden concept. How is this determined in this work and previous work it has been based on?
- Why is reasoning only important for math?
- How many samples are being filtered out of the answers are not correct (L222)?
- Not a weakness: As the paper has adopted the same architectures as previous work, did the authors try more capable/recent models?

---

### Official Review · Reviewer_rPTC · 2025-10-30

**Soundness:** 2
**Presentation:** 2
**Contribution:** 2
**Rating:** 2
**Confidence:** 3

**Summary:**

The paper studies composing a multilingual model with a reasoning-oriented LLM (a “merger” framework). It aims to adaptively balance contributions from the multilingual encoder/representations and the reasoning model by training lightweight residual adapters with a preference-optimization objective (DPO) plus an NLL loss term. The method targets better trade-offs across high-resource and low-resource languages and evaluates on multilingual reasoning/understanding benchmarks.

**Strengths:**

* Important problem: Composition/merging of multilingual and reasoning abilities is an impactful direction, and the paper provides empirical results demonstrating consistent (if modest) gains on multiple multilingual tasks.
* Lightweight & implementable: The approach freezes backbones and only trains small adapters/coefficients with standard losses (DPO + NLL), making it easy to integrate into existing pipelines.
* Comprehensive analysis: The paper includes correlation analyses, ablations (e.g., removing NLL or iterations), and qualitative diagnostics (e.g., cross-language consistency), which help interpret where the gains come from.

**Weaknesses:**

1. Limited novelty: At its core, the method reduces to training small adapters on a composed model using standard DPO (with an auxiliary NLL). The “adaptive gate” is largely conceptual—there is no explicit gating module, and the work does not introduce a fundamentally new optimization or architectural mechanism.
2. Loose method–motivation alignment: The paper motivates an input-adaptive gating to balance multilingual vs. reasoning signals, but the implementation does not expose or analyze an explicit gate (granularity, parameterization, routing behavior). It remains unclear whether DPO-trained adapters truly realize the hypothesized gating dynamics beyond generic fine-tuning effects.
3. Clarity issues:
   - The description of the “gate/coefficients” is ambiguous (global vs. layer-wise vs. head-wise; conditioned on language ID or not; token- vs. sequence-level).
   - The construction of DPO triplets is under-specified: What exactly is the query distribution, and what is the reference policy in DPO?

4. Generalization & robustness under-evaluated: Results are shown for one primary composition pair, lacking of results on other model pairs.

5. Efficiency trade-offs (latency/memory at inference) vs. baseline mergers.

**Questions:**

What is the exact parameterization of the “coefficients/gate”?

 How are DPO triplets formed concretely? What are the queries (datasets/subsets), and how are wins/losses adjudicated per task? What is the reference model selected?

---

### Note · Authors · 2025-11-18

I have read and agree with the venue's withdrawal policy on behalf of myself and my co-authors.